

# Processing causal structure sentences in Mandarin Chinese: an eye movements study

Lei Gao[1], Lin Li[2,3], Xiaolei Gao[1] and Xue Sui[3]

[1] School of Education, Tibet University, Lhasa, Tibet, China
[2] Department of Psychology, Tsinghua University, Beijing, China
[3] School of Psychology, Liaoning Normal University, Dalian, China

## ABSTRACT

It remains uncertain whether causal structure prediction can improve comprehension in Chinese sentences and whether the position of the headword mediates the prediction effect. We conducted an experiment to explore the effect of causal prediction and headword position in Chinese sentence reading. Participants were asked to read sentences containing causal connectives with their eye movements recorded. In the experiment, we manipulated the causal structure of the sentence and the position of the headword. We found a promoting effect of causal structure on first-pass reading time and a hindering impact on total reading time. However, the effect was not mediated by the headword position. The results show that causal syntactic prediction facilitated early-stage processing and increased the integration cost in the late stage of Chinese sentence processing. These findings also support the constraint-based approach, which suggests an isolation between semantic and syntactic processing.

## INTRODUCTION

During human communication, sentences are often connected by various connectives, such as paratactic conjunctions, selective conjunctions, and causal conjunctions. This study focuses on causal conjunctions, which establish causal relations between sentences (*Canestrelli, Mak & Sanders, 2013*), with the former being the cause and the latter being the effect. Conjunctions signal coherence relations between discourse units, with readers prompted to expect explanations for preceding statements and to integrate upcoming statements into a coherent mental representation (*Hussein, 2008*; *Murray, 1995*).

Conjunctions always play a predictive role. When reading a text, readers encounter causal connectives and anticipate the causal relationship between the preceding content and the following one. This anticipation influences how the preceding contents affect the processing of the following content (*Ferreira, 1996*; *Frazier, Munn & Clifton, 2000*). Previous studies have pointed out that prediction is a complex interplay between top-down and bottom-up influences on perception and action (*Clark, 2013*). Eye movement, electrophysiological, and functional imaging studies provided evidence for the existence of

Corresponding authors
Lin Li, tessa928@163.com
Xue Sui, suixue88@163.com

the prediction effect in language processing and production (*Baus et al., 2014*; *Shao et al., 2022*; *Li, Li & Qu, 2022*). However, does causal sentence processing always benefit from prediction? The aim of the present study is to investigate the prediction effect in causal compound sentences.

*Gibson (1998)* used the dependency locality theory (DLT) to explain the mechanism of sentence processing, arguing that sentence processing involves two costs: storage and integration costs. Retaining the predicted contents in working memory consumes storage costs, also called memory costs. The integration of expected content with context incurs integration costs. *Gibson (2000)* claims that information predicted by previous content must be stored during reading, leading to slower subsequent language processing. *Chen, Gibson & Wolf (2005)* conducted experiments to test for the effects of storage cost through the manipulation of the number of predicted verbs, as shown in the following examples in (1):

(1) a. Zero predicted verbs (Verb 1/Verb 2): The detective suspected that the thief knew that **the guard protected the jewels** and so he reported immediately to the museum curator.

b. One late predicted verb (Verb 1/Noun 2): The detective suspected that the knowledge that **the guard protected the jewels** came from an insider.

c. One early predicted verb (Noun 1/Verb 2): The suspicion that the thief knew that **the guard protected the jewels** worried the museum curator.

d. Two predicted verbs (Noun 1/Noun 2): The suspicion that the knowledge that **the guard protected the jewels** came from an insider worried the museum curator.

They found that reading times increased with the number of predicted verbs. They hypothesized that the prediction of syntactic structure (which involves information being retained in working memory) increases memory costs and hinders comprehension. *Grodner & Gibson (2005)* experiment also verified the existence of storage costs. Their studies suggested that the integration of a new input item poses a challenge, largely depending on the amount of lexical material between the input item and its intended dependents. From a resource-limited perspective of language processing, lengthy integrations create difficulty even in unambiguous sentences.

Conversely, *Ferreira (1996)* proposed that the discourse is gradually established. When inputting linguistic components, readers analyze possible syntactic structures, with the most probable configuration selected and activated. This activated syntax can facilitate understanding sentences when subsequent content is consistent with predictions. Similarly, *Frazier, Munn & Clifton (2000)* found that similar syntax benefited sentence processing by reducing both FDT and total reading time for adverbs or prepositional phrases. *Staub & Clifton (2006)* also observed that syntactic markers prevented incorrect analyses of noun phrase coordination and thus reduced first-pass and go-past reading times for subsequent noun phrases. They suggested that the connective words activate the syntactic parser and facilitate syntactic construction. Overall, many studies highlight the

importance of syntactic structure and prediction in sentence comprehension and reading efficiency.

In summary, conflicting hypotheses are currently presented regarding the structural effects–either facilitating (*Frazier, Munn & Clifton, 2000*; *Kliegl et al., 2004*) or hindering processing (*Gibson, 2000*; *Chen, Gibson & Wolf, 2005*). This inconsistency may imply different processing mechanisms: prediction and integration. According to *Pickering & Gambi (2018)*, prediction in language comprehension involves the activation of linguistic information before encountering the input that carries that information. In contrast, integration occurs when a reader combines activated linguistic information with the representation of preceding input (*i.e.*, the context). Integration does not involve pre-activation and, therefore, does not facilitate comprehension in the same direct manner as prediction.

As an illustration of *Staub*'s *(2010)* study, object clauses were found to be processed more difficultly than subject clauses. Importantly, language processing is likely to be more difficult when the headword is farther back in the sentence (*Gibson, 1998*; *Grodner & Gibson, 2005*). These studies emphasize that the proximity (or locality) of the sentence's elements predicts online comprehension difficulty, with long-distance dependencies leading to integration problems. This suggests that memory retrieval (*Martin & McElree, 2008*; *McElree, Foraker & Dyer, 2003*) and structural prediction (*Staub & Clifton, 2006*; *Lau et al., 2006*) are involved in syntactic processing. Therefore, according to Locality Theory, the causal syntactic effect might also be influenced by the position of the headword (e.e., the distance of integration).

A headword is a phrase (a compound word or a phrase) interpreted as a label for a set of concepts; sentences are built and processed around a headword (*Uchida & Zhu, 2001*). In the current study, the headword represents the central information, modified and restrained by modifiers and syntax. All the headwords used in the current study were noun phrases. Researchers argue that the later the "headword" (information predicted by the previous context) appears, the more processing resources it consumes. This hypothetical influence of headword position on the costs of sentence processing is referred to as the headword position effect (*Kliegl et al., 2004*; *Gleitman et al., 2007*). The prediction effect of language processing is influenced by syntactic structure (controlled by causal connectives in the current study) and headword position.

To investigate sentence structure and position effects, we chose causal connective as one of the independent variables. According to previous studies (*Zhang & Mo, 2003*; *Zeng, 2016*), the causal compound sentence is one of the most common sentence patterns in Chinese. In Chinese, connectives "因为" and "所以" (translated as "because" and "so") can appear together in a sentence. In the experiment, we used causal compound sentences as experimental materials to investigate the predictive effect. DLT claims that processing difficulty is related to the distance of the headword (noun phrase), so we aim to determine whether the distance of connective also impacts sentence processing. We set three levels of causal connectives: no-connective, half connectives (…所以…), and full connectives (因为…所以…). Besides syntactic structure predictions, readers also make inferences based on their knowledge and experience, with this contribution from top-down
processing expected across conditions. In our study, we evaluate the predictive effect of syntactic structure and control for the headword position (former position or latter position) to test whether the prediction effect is affected by the position of the headword in the sentence.

We hypothesized that causal connectives would decrease the subsequent processing difficulty of sentences, thus cutting reading times. The rationale behind this hypothesis is that causal connectives activate related information in advance, thereby facilitating later processing. However, due to the increased storage cost when headword is behind, it is also possible that more regressions to the first clause could be observed under causal connective conditions.

Additionally, the integration cost increases when the headword is placed at the end of the sentence. Therefore, readers were assumed to spend more reading time on early and late-stage measures under postpositional headword conditions. Furthermore, according to locality theory, if the headword placement can moderate the causal connective facilitatory effect, readers will spend more reading time under the final headword and no connective condition.

## MATERIALS AND METHODS

### Participants

Forty Chinese native speakers (29 women, aged 19–21 years) were recruited from the university community and were rewarded with a small gift for their participation. All participants had normal or corrected-to-normal visual acuity, with no history of any reading disorder. The study was approved by the Ethics Committee of Tibet university (XZDXLL2022016), with informed consent provided to all participants.

To assess the statistical power of our experiment, we conducted a power analysis using the simr package, with a focus on the fixed effects: sentence structure, headword position, and their interaction. At a 95% confidence interval, the estimated power was 0.9 for sentence structure and 0.2 for both headword position and the interaction effect. These results indicate that, given the current sample size and model settings, we have sufficient powers to detect the significance of the fixed and interactive effects.

### Apparatus

Sentences in black type were displayed in a single line on a light gray background, one sentence at a time. Experiment Builder (version 2.2.1) was used to display the sentences. A 17.7-inch CRT monitor running at 120 Hz presented Song font text at a display resolution of $1,024 \times 768$ pixels. The distance between the reader's eyes and the monitor was approximately 70 cm. At this distance, 0.8 characters subtended about $1°$ of visual angle laterally. Viewing was binocular, but eye movements were recorded only from the right eye using an Eyelink 1000 (SR Research, Mississauga, Ontario, Canada) running at a sampling rate of 1,000 Hz. The participant's head was positioned on a chin rest to minimize head movements. The relative average accuracy of measurement was in the order of $0.15°$, whereas absolute accuracy was maintained at less than one-character width *via* calibration and validation routines (*Inhoff & Radach, 1998*).

## Materials and design

A total of 36 target sentence frames were used. Headwords (compound words) were controlled for common word frequency and similar complexity (average stoke count: M = 7.88, SD = 1.63). Common frequency headwords were defined as those appearing more than 10,000 times and less than 35,000 times in the BCC *corpus* (*Xun et al., 2016*). Each headword consisted of 3-5 characters (M = 3.7, SD = 0.55). Each target sentence consisted of 24–31 characters, with two clauses connected by causal connectives. Every sentence was assessed to ensure grammatical and semantic accuracy. A norming study was conducted with 26 participants who did not subsequently attend the eye movement experiment. Participants rated the extent of the causal relationship between two clauses without any causal connective on a scale of 1 to 5, where one denoted "completely without causal relationship," and five indicated "a strong causal relationship." The mean score was 4.00 (range: 3.54–4.65). After this evaluation, the same participants assessed the naturalness of the sentences, with all sentences judged to be adequately natural (*M* = 4.42). Conditions did not differ significantly (*p* > 0.05).

The study employed a 3 × 2 factorial design (see Table 1). The most common causal connective ("因为…, 所以…") was used, with three levels of the causal connective condition: no connective condition, half connective condition ("……, 所以……"), and full connective condition ("因为…, 所以…"). All connectives were positioned at the beginning of the respective clauses. Headword position was manipulated across two levels: at the beginning of the clause and at the end of the clause. It was important that swapping the headword position not alter the meaning or coherence of the sentences.

The sentences were counterbalanced across conditions and presented in a randomized order to each participant. Additionally, 54 filler sentences without causal relationships were included. Among these, twelve sentences had no connectives, while other sentences featured connectives, such as either… or…, if… then…, not only… but also… and although… (but)… All sentences were grammatically correct in Chinese. Each participant read only one of the six lists provided.

## Procedure

All participants completed the experiment individually. Before the trials, the accuracy of the monitoring system was checked and re-calibrated (if necessary). Participants were required to read each sentence and then either pressed a button to continue or responded to yes-no comprehension questions presented after approximately 40% of the trials. Three practice trials were presented at the start of the session. The entire experiment lasted approximately 30 min.

## RESULTS

The accuracy rate of reading comprehension was 95.7%, indicating that participants read carefully and understood the sentence content. Eye-movement data were extracted using Data Viewer (version 3.1.264). Trials were excluded if tracker loss occurred, or if fixation durations were less than 80 ms or greater than 1,200 ms. Additionally, eye movement

**Table 1  Sample of target sentences with the manipulated factors of causal connective conditions and the headword position.**

| Condition (causal structural condition/headword position) | First clause | Second clause |
|---|---|---|
| No connective/prepositional headword | 路面上积雪很深,<br>There is very deep snow on the road, | *出行不便*是最大的问题.<br>*the traffic inconvenience* is the biggest problem. |
| No connective/postpositional headword | 路面上积雪很深,<br>There is very deep snow on the road, | 最大的问题是*出行不便*.<br>the biggest problem is *the traffic inconvenience*. |
| Half connective/prepositional headword | 路面上积雪很深,<br>There is very deep snow on the road, | 所以*出行不便*是最大的问题.<br>(so) *the traffic inconvenience* is the biggest problem. |
| Half connective/postpositional headword | 路面上积雪很深,<br>There is very deep snow on the road, | 所以最大的问题是*出行不便*.<br>(so) the biggest problem is *the traffic inconvenience*. |
| Full connective/prepositional headword | 因为路面上积雪很深,<br>Because there is very deep snow on the road, | 所以最大的问题是*出行不便*.<br>(so) the biggest problem is *the traffic inconvenience*. |
| Full connective/postpositional headword | 因为路面上积雪很深,<br>Because there is very deep snow on the road, | 所以*出行不便*是最大的问题.<br>(so) *the traffic inconvenience* is the biggest problem. |

**Note:**
The headword is italicized.
Underlined words are causal connectives. There was no underlining in the formal experiment.

measures that were above or below three standard deviations from the mean were excluded. In total, 2.1% of the data were removed before analysis.

Two regions were analyzed in each sentence: the second clause (clause region) and the headword (*N* region). The *Clause Region* consisted of the second clause area excluding any connective or headword, while the *N* Region consisted of the headword area in the second clause. Linear mixed models (LMMs) were run using the lmerTest package (version 3.1-0) (*Kuznetsova, Brockhoff & Christensen, 2017*) within the R environment. Causal connectives and headword position were the fixed factors for each model, and subjects and sentences were treated as random variables. The random-effects structure of the maximal model was: (1 + Causal Structure * Position | subject) + (1 + Causal Structure * Position | item). Initially, the maximal model was fitted to the data, but some of the random-effects structures failed to converge. Appropriate random-effects structures were then established for each analysis (see Table 2). FDR adjustments were used to compare causal structures and headword positions. All findings reported here are thus from successfully converging models.

Four measures were analyzed in the clause region: first pass reading time (FPR), the fixation count of first pass (FC), count of regression-out (RO), and total reading time (TRT). FPR is the entire fixation duration from first entering the region until leaving the area. Both FPR and FC are related to early-stage processing (*Reichle, Rayner & Pollatsek, 2003*; *Clifton, Staub & Rayner, 2007*), while RO is to sentence integration and indicates late processing (*Rayner & Frazier, 1987*). Descriptive statistics for all reported measures are presented in Table 3.

In the N region, four measures were computed: first fixation duration (FFD), FPR, FC, and TRT. The means and standard deviations for these measures are presented in Table 4.

**Table 2  Results of the linear mixed effects model for the measures on the target across conditions with FDR adjustment.**

| IA | Measure | | Condition | P | t | SE | β |
|---|---|---|---|---|---|---|---|
| Clause region | FPR | ~C*P + (1 + position\|sub) + (1 + C*P\|trial) | Causal structure | **<0.01** | −3.43 | 55.47 | −19.18 |
| | | | Position of headword | 0.46 | −0.74 | 26.01 | −19.17 |
| | | | C*P | 0.66 | −0.43 | 110.96 | −48.18 |
| | FC | ~C*P + (1 + position\|sub) + (1 + C*P\|trial) | Causal structure | **0.02** | −2.53 | 0.21 | −0.52 |
| | | | Position of headword | 0.37 | −0.89 | 0.1 | −0.09 |
| | | | C*P | 0.8 | 0.32 | 0.41 | −0.1 |
| | RO | ~C*P + (1 + C*P\|sub) + (1 + C*P\|trial) | Causal structure | **0.04** | 2.07 | 0.06 | 0.12 |
| | | | Position of headword | 0.95 | 0.06 | 0.03 | 0.002 |
| | | | C*P | 0.83 | 0.22 | 0.11 | 0.02 |
| | TRT | ~C*P + (1 + position\|sub) + (1 + position\|trial) | Causal structure | **<0.01** | 5.38 | 43.07 | 27.3 |
| | | | Position of headword | 0.99 | 0.0004 | 42.62 | −12.72 |
| | | | C*P | 0.91 | 0.09 | 60.1 | 10.69 |
| N region | FFD | ~C*P + (1 + C*P\|sub) + (1 + C*P\|trial) | Causal structure | **<0.01** | −4.65 | 5.56 | −21.97 |
| | | | Position of headword | **<0.01** | −4.81 | 4.92 | −23.66 |
| | | | C*P | 0.14 | −1.46 | 20.83 | −30.44 |
| | GD(FPR) | ~C*P + (1 + position\|sub) + (1 + C*P\|trial) | Causal structure | **0.02** | −2.5 | 25.63 | −64.07 |
| | | | Position of headword | 0.69 | −0.4 | 12.14 | −4.89 |
| | | | C*P | 0.92 | 0.1 | 51.31 | 5.23 |
| | FC | ~C*P + (1\|sub) + (1\|trial) | Causal structure | 0.05 | 0.09 | −1.93 | −0.17 |
| | | | Position of headword | **<0.01** | 4.52 | 0.04 | 0.19 |
| | | | C*P | 0.52 | 0.65 | 0.18 | 0.12 |
| | TRT | ~C*P + (1 + C*P\|sub) + (1 + C*P\|trial) | Causal structure | **0.02** | −2.63 | 31.83 | −83.58 |
| | | | Position of headword | **<0.01** | 4.22 | 15.07 | 63.54 |
| | | | C*P | 0.62 | 0.49 | 63.71 | 31.33 |

**Notes:**
Significant P values are in bold. B referred to the estimate values of each condition.
C*P: Causal structure*Position of headword.

**Table 3  Mean reading measures of Clause region for each condition for the Experiment (unit: millisecond).**

| Measures | Prepositional headword | | | Postpositional headword | | |
|---|---|---|---|---|---|---|
| | No connective | Half connective | Full connective | No connective | Half connective | Full connective |
| FPR | 1,338 (53) | 1,156 (65) | 1,117 (58) | 1,359 (61.29) | 1,143 (64) | 1,168 (62) |
| FC | 5.04 (0.21) | 4.49 (0.17) | 4.39 (0.19) | 5.1 (0.21) | 4.45 (0.2) | 4.52 (0.21) |
| RO | 0.26 (0.04) | 0.4 (0.03) | 0.39 (0.04) | 0.27 (0.03) | 0.39 (0.04) | 0.38 (0.03) |
| TRT | 1,335 (37) | 1,383 (40) | 1,421 (44) | 1,325 (41) | 1,365 (46) | 1,439 (43) |

**Note:**
Parentheses contain the standard error.

## Causal syntactic structure

### Clause region

The FPR for the half-connective and full-connective conditions significantly differed from the no-connective condition ($t = -3.429$, $SE = 55.468$, $\beta = -19.18$, $p < 0.01$), with no

**Table 4 Mean reading measures of N region for each condition for the Experiment (unit: millisecond).**

| Measures | Prepositional headword | | | Postpositional headword | | |
| --- | --- | --- | --- | --- | --- | --- |
| | No connective | Half connective | Full connective | No connective | Half connective | Full connective |
| FFD | 266 (6) | 248 (9) | 246 (5) | 278 (9) | 273 (8) | 279 (9) |
| GD(FPR) | 489 (21) | 448 (24) | 433 (21) | 473 (23) | 492 (29) | 449 (27) |
| FC | 1.91 (0.01) | 1.83 (0.03) | 1.8 (0.03) | 1.68 (0.04) | 1.79 (0.05) | 1.62 (0.04) |
| TRT | 630 (34) | 624 (55) | 588 (27) | 577 (68) | 587 (32) | 535 (73) |

Note:
Parentheses contain the standard error.

significant difference between the half-connective and full-connective conditions ($p > 0.05$). This indicates that the presence of connectives (whether full or half) aids in the initial reading process compared to the absence of connectives. However, the extent of connectivity (full *vs.* half) does not impact FPR.

A reliable effect of the causal structure was found for FC ($t = -2.528$, $SE = 0.206$, $\beta = -0.52$, $p < 0.05$), with the half-connective and full-connective conditions significantly different from the no-connective condition ($p < 0.01$) but not from each other. More fixation counts were found in the initial reading period for the no-connective condition than for the other two conditions. This suggests that the absence of connectives results in more fixations during the initial reading, indicating that connectives (both half and full) facilitate smoother reading with fewer fixations.

The effect of the syntactic structure was reliable for the RO ($t = 2.066$, $SE = 0.056$, $\beta = 0.12$, $p < 0.05$). The RO sequentially decreased from the half connective condition, through the full connective condition, to the no connective condition. The no connective condition significantly differed from the half connective and full connective conditions ($p < 0.05$), but there was no significant difference between the latter two conditions. This suggests that the presence of connectives (both half and full) reduces the need for regressions (re-reading), facilitating the comprehension and processing of text. The no connective condition requires more regressions, indicating difficulties in understanding or processing.

The effect of the syntactic structure was reliable for the TRT ($t = 5.382$, $SE = 43.07$, $\beta = 27.3$, $p < 0.05$). The TRT of the full-connective condition was shorter than that of the no-connective condition ($p < 0.05$) and the half connective level ($p < 0.05$). Full connectives most effectively reduce the total reading time, indicating that they provide the greatest facilitation for reading and comprehension. While half connectives are helpful, they are not as effective as full connectives in reducing total reading time compared to the no-connective condition.

The presence of connectives (both full and half) significantly improves the reading process by reducing FPR, fixation count, and the need for regressions, as well as by decreasing the total reading time. Full connectives are particularly effective in reducing the total reading time, highlighting their importance in facilitating reading and comprehension. The no-connective condition poses more challenges for readers, resulting in longer reading times, more fixations, and more regressions.

### N region

FFDs showed a significant effect of syntactic structure ($t = -4.647$, $SE = 5.562$, $\beta = -21.97$, $p < 0.001$). Values for the no-connective condition significantly differed from the half-connective and full-connective conditions ($p < 0.001$), but the latter two conditions did not significantly differ from each other.

The FPR of the half-connective and full-connective conditions was shorter than for the no-connective condition ($t = -2.499$, $SE = 25.632$, $\beta = -64.07$, $p < 0.05$).

A significant effect of syntactic structure was found for FC ($t = -1.934$, $SE = 0.089$, $\beta = -0.17$, $p = 0.05$), with smaller values for the half-connective and full connective conditions compared to the no-connective condition.

TRT was shorter for the full-connective condition compared to the other two conditions ($t = -2.626$, $SE = 31.829$, $\beta = -83.58$, $p < 0.01$).

The presence of connectives, particularly full-connectives, improves reading efficiency by reducing initial fixation durations, the number of fixations needed, and overall reading time. This indicates that syntactic structures provided by connectives significantly facilitate sentence comprehension during reading.

## Position effect

### Clause region

Headword position produced no significant effect, and there was no interaction between causal structure and headword position ($ps > 0.05$).

### N region

FFDs showed a significant effect of syntactic structure ($t = -4.647$, $SE = 5.562$, $\beta = -21.97$, $p < 0.001$). The no-connective condition significantly differed from both the half-connective and full-connective conditions ($p < 0.001$), but the latter two conditions did not significantly differ from each other.

FC for the prepositional condition was higher than for the postpositional condition ($t = 4.523$, $SE = 0.042$, $\beta = 0.19$, $p < 0.01$).

TRT for the postpositional condition was shorter than for the prepositional condition ($t = 4.216$, $SE = 15.071$, $\beta = 63.54$, $p < 0.01$).

### Interactive effects

No other effect was found for this measure ($ps > 0.05$).

## DISCUSSION

This study explored the influence of causal structure during Chinese sentence reading. The results show that causal syntactic structure plays an important role in both early-stage and late-stage processing. The position of the headword does not influence the role of the causal syntactic structure. Consistent with previous research findings (*Millis & Just, 1994*), there was no difference between the two conditions that included causal connectives, both of which showed FPR than the no-connective conditions in the N and Clause regions. The trend of fixation counts in the first reading is consistent with findings for FPR, with the increase in FPR synchronized with a rise in fixation counts. The results show that syntactic

structure prediction plays a significant role in early-stage processing; causal syntactic structure reduces the difficulty of lexical access in the early stage. The prediction effect of the syntactic structure is not affected by the headword position; rather, this position specifically affects the processing of the headword itself.

### Influences of causal structure on sentence processing

Causal prediction occurs in the early-stage and late-stage processing. We found more total reading time under connective conditions than under no-connective conditions due to regressions in the Clause region analyses. This process shows that the structure parser is actively building the relevant syntax to facilitate early lexical access (*Staub & Clifton, 2006*), which in turn slows down processing in the late stage. Unlike the early-stage processing findings, we found more total reading time was observed under connective conditions than under no-connective conditions due to regressions in the Clause region analyses. In line with locality theory, syntactic structure increases the integration cost of sentence processing, resulting in longer reading time during late-stage processing. Longer FPR was found for the postpositional than the prepositional headword condition, while longer total reading time was found for the prepositional than the postpositional condition.

In the absence of causal connectives, structural representation is hardly activated, and no competition needs to be solved or integrated during late-stage processing. Consistent with the views of *Gibson (1998)*, we found that syntactic structure hinders sentence processing during the late stage. More integration time in connective conditions suggests an isolation or different processing mechanism between semantic and syntactic processing.

### Position effect in Chinese causal sentence

In previous research, readers appeared to take longer to process the headword when it is postpositional due to the wrap-up effect (*Warren, White & Reichle, 2009*). However, in our experiment, the total reading time was lower when the headword was postpositional, which can be explained by two reasons: First, no subject preference was found in the Chinese language. Second, the processing of the last word included both syntactic and semantic prediction; perhaps, in our materials, readers could easily predict the content at the final position. The current study indicated that causal syntax integration occurred in late-stage processing. According to DLT (*Demberg & Keller, 2008*), integration cost can be regarded as a backward-looking cost that causes a delayed effect. Therefore, the immediate effect may be caused by the wrap-up, while the delayed effect by sentence integration.

### Interaction

No interaction between causal structure and headword position was found in the current study. One explanation is that the impact of connectives is more significant than the storage cost assumed to remain. The headword is activated by the causal connectives regardless of its position. Another possible explanation is the absence of inherent position preference for the headword, and the probabilities of it being a subject or an object are similar (*Hsiao, 2003*; *Qiao, Shen & Forster, 2012*). Therefore, the causal prediction effect was the same in both prepositional and postpositional headword conditions.

In previous studies, very complex or ambiguous clauses in nested sentence structures were mostly used. In contrast, the clauses used in this study were all simple SVO structures. This SVO structure conforms to typical Chinese text, which mainly consists of simple and short sentences, with few complex modifications or nested structures. The proportion of subject clause and object clause is almost the same (57.5% and 42.5% respectively), indicating no position preference for headwords (*Hsiao, 2003*).

## CONCLUSIONS

This study confirms the idea that the predictive effect of causal syntax can both facilitate and hinder language processing. Additionally, the position of the headword does not alter the causal prediction effect. The current study focused exclusively on one syntactic structure processing; Further studies should explore a wider variety of structures to delve deeper into the underlying sentence processing mechanism. Moreover, an investigation into children's processing patterns could reveal how the syntactic prediction mechanism develops.

## ACKNOWLEDGEMENTS

The authors gratefully acknowledge the subjects who participated in the study and Chen Xinyi for data collection.

### Funding

This research was supported by the National Natural Science Foundation of China in 2022 (32260204) funded to Xiaolei Gao. The funders had no role in study design, data collection and analysis, decision to publish, or preparation of the manuscript.

### Grant Disclosures

The following grant information was disclosed by the authors:
National Natural Science Foundation of China in 2022: 32260204.

### Competing Interests

The authors declare that they have no competing interests.

### Author Contributions

- Lei Gao conceived and designed the experiments, performed the experiments, prepared figures and/or tables, and approved the final draft.
- Lin Li conceived and designed the experiments, performed the experiments, analyzed the data, prepared figures and/or tables, and approved the final draft.
- Xiaolei Gao analyzed the data, authored or reviewed drafts of the article, and approved the final draft.
- Xue Sui conceived and designed the experiments, authored or reviewed drafts of the article, and approved the final draft.

## Human Ethics

The following information was supplied relating to ethical approvals (*i.e.*, approving body and any reference numbers):

The study was conducted in accordance with the Declaration of Helsinki and approved by the Institutional Review Board of Tibet University (XZDXLL2022016).

## Data Availability

Code and data are available at OSF:

Li, Lin. 2023. "Causal Prediction Effect." OSF. November 10. osf.io/u5fps.

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
