# Peer review of "Processing causal structure sentences in Mandarin Chinese: an eye movements study"

_PeerJ, doi:10.7717/peerj.17878_

## Round 0.1 · original submission · Major Revisions

· Academic Editor

Major Revisions

After looking over the reviewer comments all comments appear reasonable, and I encourage the authors to go through them all and make changes to improve the manuscript. If there are any reviewer comments the authors disagree with and do not make edits, please provide a rationale in response for why changes are not made.

One additional minor comment from myself, if the authors would like the paper to be more accessible to a wider audience I would encourage including more example sentences to help complement the text like what was done lines 96-109. I can see quite a few places throughout the introduction and discussion sections where such examples would be a useful complement to the text. Using some full sentence examples would be useful as well, I expect. I make this suggestion for you to take or leave as you see fit.

Reviewer 1 ·

Basic reporting

The manuscript is generally well-written. The Introduction is a little terse in places and might benefit from expanding on the discussion of key studies to describe what they did.

The measures reported for the headword region need to be defined.

The authors describe effects in terms of being predictive. However, these effects are as likely, and possibly more likely, to reflect integration effects. The authors should be clearer about the distinction between prediction and integration (I suggest looking at Pickering and Bambi, Psychological Bulletin).

The authors talk about the increased total time effects for "predictive" as hindering processing. It's not clear to me that this is the case but they should explain the underlying mechanisms. What is being hindered?

Experimental design

It would be helpful to say more about the decisions regarding sample size and stimulus size. There were only 36 stimuli across 6 experimental condition (3x2 design). This is only 6 observations per condition, which seems rather low. Ideally, the research might address this issue by conducting a power sensitivity analysis to estimate the minimum effect size that the study is powered to detect, and to compare this with the observation effect sizes.

Line 110-116. Who is they in this paragraph? Greater clarity is needed about the specific hypotheses to be tested in this experiment. If you are testing the hypotheses attributed to "they" please make this clear.

Validity of the findings

It's difficult to follow the experimental findings. This might be clearer if they results are reported by each independent variable rather than by each eye movement measure. This would allow the researchers to describe the pattern of effects produced by each independent variable. It would be helpful to comment in the Results section about the relationship between the findings and the tested hypotheses.

Additional comments

it's unclear if any of the statistical models converged. It would be helpful to report which models were used for each measure - was the model reported at lines 188-196 used for all measures. (Note also the misspelling of trial at line 195).

Reviewer 2 ·

Basic reporting

The writing is, in general, clear and unambiguous. However, there are many instances where words / expressions are used incorrectly. I will refer to many of these instance below. I am not a native English-speaker myself. A thorough language editing from a fluent English speaker would benefit the manuscript.

In the introduction, the authors do an excellent job in providing the context and the background. I am not an expert in this matter (linguistics), but had little difficulty to understand the rationale of the study.

The structure of the manuscript conforms to the PeerJ standards with regard to the main headings. The PeerJ standard for subheadings are: “Subheadings must be bold, followed by a period, and start a new paragraph.“ This, the authors did not follow, but this is easily remedied.

The manuscript does not contain figures. The tables are are informative and properly formatted. I may advice the authors not to report decimals, if the measure is in milliseconds. (In my opinion, fractions of milliseconds are almost infinitesimal small and hence uninformative and irrelevant.)

The raw data is supplied in an OSF repository. This repository is accessible via a link in the submission forum. In the manuscript itself, however, there is no information about the accessibility of the data and the link is not provided (unless I missed it).

Experimental design

“PeerJ [...] considers articles in the Biological Sciences, Environmental Sciences, Medical Sciences, and Health Sciences.” Strictly speaking, the research reported in the manuscript is not within the scope of the journal.

The research question of the manuscript is well defined, relevant and meaningful. The authors clearly stated the knowledge gap and the findings of the study fills it.

The methods are of a high technical standard and the investigation / experimental design is rigorous.

The methods are described with sufficient detail and the information is sufficient to replicate the study.

Validity of the findings

The manuscript fulfills the criteria that the conclusions are well stated, linked to the original research and limited to supporting results.

Additional comments

These are language issues / wording issues I spotted. The list is not complete. Professional language editing is highly recommended.

Line 26 (Abstract). The “Still” would better be a “However”.

The last sentence of the Abstract should be in presence (“supports” rather than “supported” and “implies” rather than “implied”).

Line 34. What does “in this piece” mean? Does it mean “the present study”?

Line 36. Add an “at” after “hints”

Line 45. Replace “proved” with “provided evidence”.

Line 55. I do not understand “reading times increased with the number of the predicted verb”. “the predicted verb” is one verb. What does “the number of” refer to? Please clarify.

Line 111. Who is “They”? Do you mean “We”? (Same issue with “their” in line 112)

Line 152. Replace “A normed study” with “A norming study”.

Line 160. Delete the second occurrence of “used”.

Line 169. Change “We’ve” to “We”.

The two sentences on lines 170 and 171 should be in the past tense (change “have” to “had”).

Line 172. Either change “grammatically” to “grammatical” or add “correct” after “grammatically”.

Line 158. Reformulate the sentence “No significant effect was also found across conditions (p >
.05)” as, e.g., “Conditions did not differ significantly (p < .05).”

In line 185f the authors state that “All eye movement measures above or below three standard deviations from the mean were excluded.” Please state explicitly whether this is the mean of the entire group of participants or the mean of the individual participant.

Line 192. “The random-effects structure of the maximal model was failed to converge.” Delete “was”.

Concerning the abbreviations (line 200ff), first pass reading time is usually abbreviated as FPR, fixation count may be abbreviated as FC (rather than FCF), regressions out as RO; TRT for total reading time is fine.

Line 255. You may change “This is not regulated by headword position.” to “The role of the causal syntactic structure is not influenced by the position of the headword.”

Line 259. Change “first reading time” to “first pass reading time”.

Line 261. Delete “vital” or replace it with “significant”.

You may delete (or reformulate) the first sentence of the paragraph starting at line 278.

Line 289. “First, no subject preference was found in the Chinese language” suffice. You may delete the remainder of the sentence.

Line 292. Delete “following”.

Line 295. Replace “derived from” with “caused by”.

Line 307. Delete “and scattered”, or explain what it means. What are “scattered” sentences?

---

## Round 0.2 · accepted · Accept

· Academic Editor

Accept

Dear authors,

Thank you for your re-submission. Your response document was thorough and enabled me to understand that you had engaged with the provided feedback and made appropriate changes. I believe the clarity of the manuscript has significantly improved after these modfications. Due to this, I don't see any need for further review.